# Historical Landscape Elements of Abandoned Foothill Villages—A Case Study of the Historical Territory of Moravia and Silesia

**Hana Vavrouchová** *[ID], **Antonín Vaishar** [ID] **and Veronika Peřinková**

Department of Applied and Landscape Ecology, Faculty of AgriSciences, Mendel University in Brno, Zemedelska 1, 61300 Brno, Czech Republic
* Correspondence: hana.vavrouchova@mendelu.cz

**Abstract:** During the second half of the 20th century, a number of settlements disappeared for various reasons, especially in the hilly landscapes of northern Moravia and in the Czech part of Silesia. Currently, in the relevant localities, it is possible to identify preserved original landscape structures (scattered greenery, water elements, original woody plants, terraces, etc.) and other historical landscape elements with heritage potential. The typical elements of the above-mentioned localities of abandoned settlements are agrarian stone walls that document previous agricultural land use. These structures are generally located outside the original building plots on the edges of previously farmed land. Another important historical element is the unused access roads to arable land, which are still visible in lidar pictures. Numerous elements of the extinct settlements also include the remains of building materials and local quarries of building stone. This paper presents and classifies the historical landscape elements and their typology and proposes a methodology for identification and documentation.

**Keywords:** landscape dynamics; historical landscape structure; abandoned settlement; cultural heritage





## 1. Introduction

During the 20th century, hundreds of settlements disappeared in the Czech Republic, with the most affected area being the border region (outside the Czech-Slovak part) and the decisive impulse being the displacement of the original German population in the post-World War II period.

These peripheral locations are made up of mountain units with less favourable climatic conditions, yet these sites have been used extensively for agriculture. These areas were mostly settled in the 13th century as part of the so-called Great Colonisation (German Eastern Colonisation) by the original Slavic and incoming German populations [1]. It was mainly German colonists who inhabited the upland and mountainous parts of the territory. The forestry, cattle breeding, mining, glassmaking, and weaving developed in these areas impacted landscape changes [2]. The settlements were conditioned by quite common factors [3]. The primary reasons for establishing settlements in these particular locations were the mineral resources (most often iron ore) and access to timber in wooded areas, namely, the higher spruce stands. Due to the widely dispersed nature of the distribution of settlements in the area, these were mostly self-contained enclaves with no significant relationships with surrounding communities (except for parish and official affiliations).

The disappearance of settlements has occurred for various reasons throughout the history of human settlement. In Central European, settlement development was completed in the Middle Ages (with very few exceptions). Since then, the number of permanent settlements has steadily decreased. The main reason for this is the progress in land management technologies and transport. This means that land can be farmed from further afield, making it unnecessary to maintain small settlements in remote locations that are difficult to

access and inefficient for modern infrastructure. This is why in some countries, villages are being depopulated as part of rural-to-urban migration [4]. However, it is possible that the situation is changing somewhat with the transition to a post-productive society, part of which may seek more remote locations with the idea of a higher quality of life [5].

Within this general trend, there are, of course, usually specific reasons for the disappearance of settlements. These causes may have been natural disasters (in the Central European environment, mainly floods; elsewhere, earthquakes, volcanic eruptions, avalanches, mudflows, etc.) or unhealthy environments [6]. Wars and violent actions, usually accompanied by economic decline, and also epidemics of infectious diseases have had a significant impact. At other times, people left for economic reasons when a territory was losing competitiveness or the local resource base was depleted.

Another reason for the disappearance of settlements is the construction of large technical works to which the settlements in their path must give way. These may be mining activities, waterworks, military facilities, or other activities. In such cases, the settlements are dismantled in a controlled manner, usually including the salvage of suitable artefacts. The affected inhabitants often protested vigorously against such action. An extreme case is the Three Gorges Dam on the Yangtze River, which resulted in the imminent displacement of at least 1.3 million people [7].

Sometimes settlements also disappear as a result of population movements due to ethnic, religious or environmental migrations. This category includes the disappearance of settlements from the territory of the present-day Czech Republic, Poland and several other countries [8,9]. Similar experiences can also be found in Poland [10,11] and Slovenia [12].

In the former Soviet Union, the disappearance of villages in the last century was associated with forced collectivisation [13]. However, the depopulation of villages in remote areas is still taking place today [14]. A study reported that about 800 villages have disappeared in Ukraine over the last 30 years [15]. However, it is not clear in how many cases the extinction is physical and when it is administrative. In Israel, villages abandoned by Arab citizens after the Arab–Israeli wars (1947) were demolished in the 1960s [16]. Research on the potential of vanished structures in Ukraine for economic restoration is also interesting [17–19].

Bański et al. [20] addressed the broader spatial and socio-economic context. Especially in the southern parts of Europe, this problem is still relevant in the context of rural and land abandonment [21]. In Bulgaria, this tendency is highlighted by the overall mass emigration of people from their country [22]. The reuse of abandoned buildings is widely discussed, e.g., [23], especially in relation to an eventual tourism function [24], for social agriculture purposes [25] or for the creation of ecovillages [26].

The study of landscapes can document the development of society and social constructs [27]. A number of studies confirm the need to interpret the links between ecosystem feedback and societal development [28]. Key changes can be seen as the transition from a pre-productive to a productive society in the past, resulting in industrialisation and urbanisation, among other things, and the transition from a productive [29] to a post-productive society in the present [30]. At present, this means that the rural landscape is changing from a space for primary production (agricultural and forestry production and mining) to a space for consumption in the context of tourism and living in a more environmentally friendly way.

For the purpose of our paper, landscape memory is one of the key concepts [31]. The mapping of landscape features for understanding landscape memory and identity is emphasised, for example, by Šťastná et al. [32]. Building on landscape history and the legacy of the past is emphasised by Biddau et al. [33]. While settlements that have disappeared in the distant past are of interest to archaeologists [34] and have become part of the historical heritage, settlements that have disappeared in the recent past touch upon identity and the present time—at least as long as there are memorials.

Crucially, as settlements disappear, so does local knowledge and socio-cultural capital [35], i.e. local culture. Sometimes some of this culture manages to be transferred to

new places; other times, it disappears almost irreversibly. It is the loss of local culture that can be seen as the main negative effect of settlement loss. Sometimes the original landscape is preserved in works of art [36]. Landscape is one of the key components of human identity and quality of life [37]. This presupposes public participation, which is in line with the spirit of the European Landscape Convention of 2000 [38]. This is the main motive of our research.

Even after several decades, a number of features can be found in the contemporary landscape that attests to the previous permanent presence of humans. These places—which can be considered traditional landscapes in terms of typology [39]—provide a good opportunity to observe the evolution of the landscape in the context of demographic change and reduced anthropic pressure on the landscape. The local landscape records a complex history of a place or region (including political decisions) that can still be read in its structure. This landscape also forms an integral part of our European cultural heritage.

Today, we are faced with the task of identifying and evaluating the changes implemented in the landscape of vanished settlements. The knowledge gained about the area can be used in landscape planning (e.g., [40,41]) and strengthening regional identity (e.g., [42]). The aim of this paper is to elaborate on a compendium and typology of landscape elements in abandoned settlements and to propose a methodology for their identification and documentation in order to provide a basis for landscape planning.

## 2. Materials and Methods

Historical landscape features were mapped on the sites of settlements that physically disappeared in the period immediately after the Second World War in the eastern part of the Czech–Polish border area (see Figure 1). The exclusive cause of the physical disappearance of settlements was the controlled demolition of buildings abandoned after the forced departure of the original German population. These sites were identified on the basis of historical demographic data and a comparison of aerial photographs from the pre- and post-World War II periods. All identified localities were involved in the study on historical landscape features.

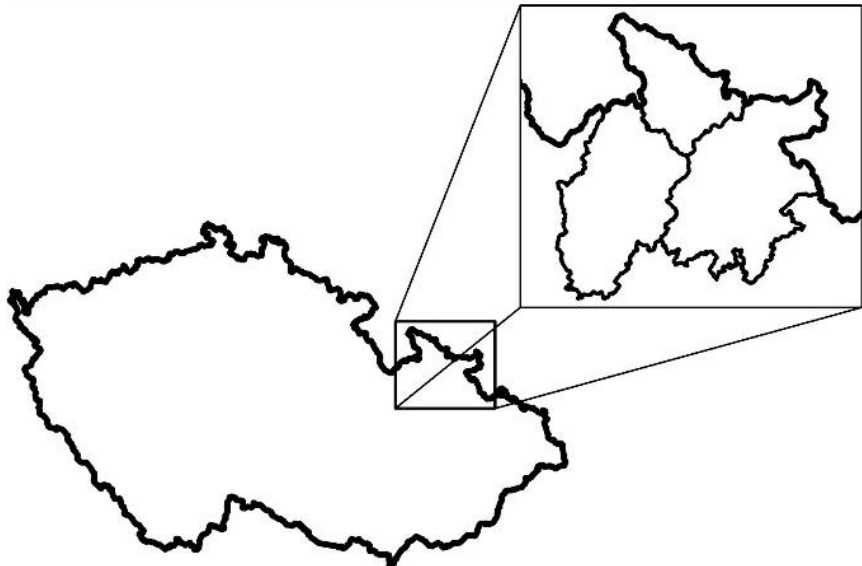

**Figure 1.** Delimitation of the addressed territory within the Czech Republic.

### 2.1. Model Area

This paper presents a narrowed area defined by the Moravian–Silesian border with Poland. The solved territory consists mainly of mountains and hilly areas (the landscape of rugged hills and the highlands of Hercinica and the landscape of distinct slopes and rocky mountain ridges, with a very rare combination of landscapes of plains and flat hills). In

terms of soil quality, less fertile soils prevail. A large part of the territory is located in a cold climate zone, characterised by a short summer. Forests and pastures predominate. Figure 2 shows a more detailed overview of the territory and location of abandoned settlements.

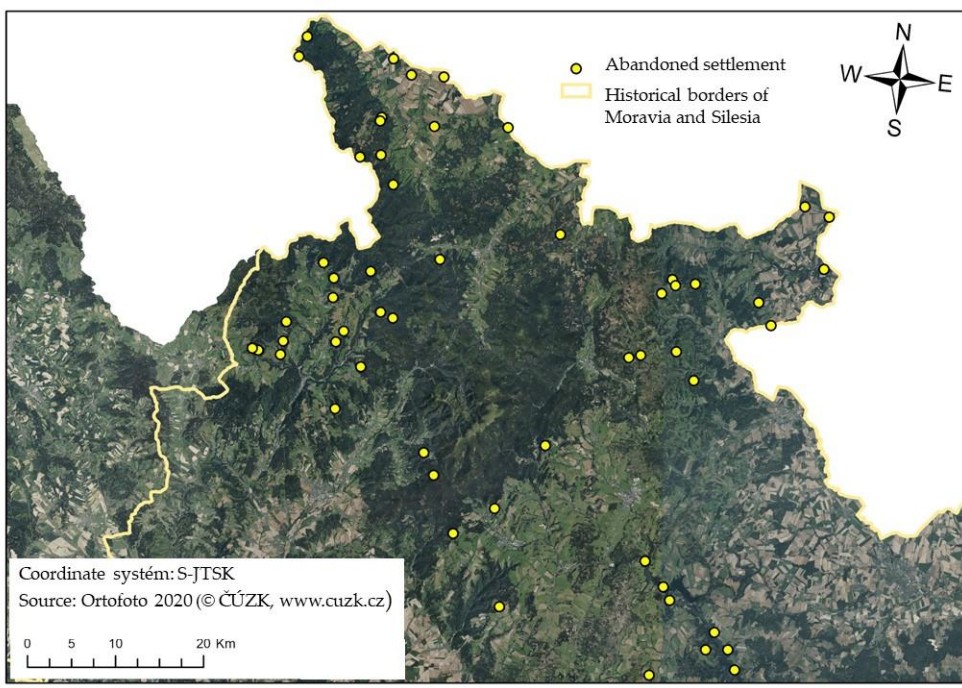

**Figure 2.** Delimitation of the addressed territory in a detailed view.

*2.2. Data Collection*

Landscape features were identified at these sites based on an analysis of current and historical aerial survey imagery (see Figure 3), a digital relief model (DRM, based on lidar scanning of the earth's surface), drone imagery and field survey. Aerial survey images and the DRM were studied within the website https://ags.cuzk.cz/archiv/ (accessed on 15 August 2021). These websites (free of charge) are provided by the Czech Office for Surveying, Mapping and Cadastre. The S-JTSK/Krovak East North coordinate system (EPSG 5514) was used. The DRM works with an absolute mean height error of 0.18 m in open landscapes and 0.3 m in forested terrains. The accuracy of DMR 5G is defined on comparative bases (152 clearly defined horizontal areas with an area of at least 50 × 50 m). Elevation point clouds are georeferenced in the UTM (Universal Transversal Mercator) coordinate reference system on the GRS 80 ellipsoid (ETRS89) and in the ellipsoidal elevation reference system relative to the GRS 80 ellipsoid. The data were collected in 2013. Surviving small (dotted), spatial and linear structures evidencing previous permanent human presence were the subject of interest, with a view to their recording, possible future conservation and use in presentation and education. The character of the landscape features and the extent of their preservation were evaluated in relation to elevation, original location in the village (intra-villan/extra-villan) and current long-term land use.

The current long-term use of the area and the extent of change from the pre-extinction landscape structure were analysed based on aerial photographs from the period immediately before (1930s–1940s) and just after the war (1950s–1960s), with the time series extended to the present. Field verification was also undertaken. The change in landscape texture was also taken into account. The structure of the landscape can be understood as the spatial distribution of landscape elements (fields, forests, settlements, etc.) connected by mutual relations. Texture is a spatial representation of the landscape structure, taking into account the size of individual homogeneous areas (the background is made up of highlighted visible lines and polygons based on aerial survey images).

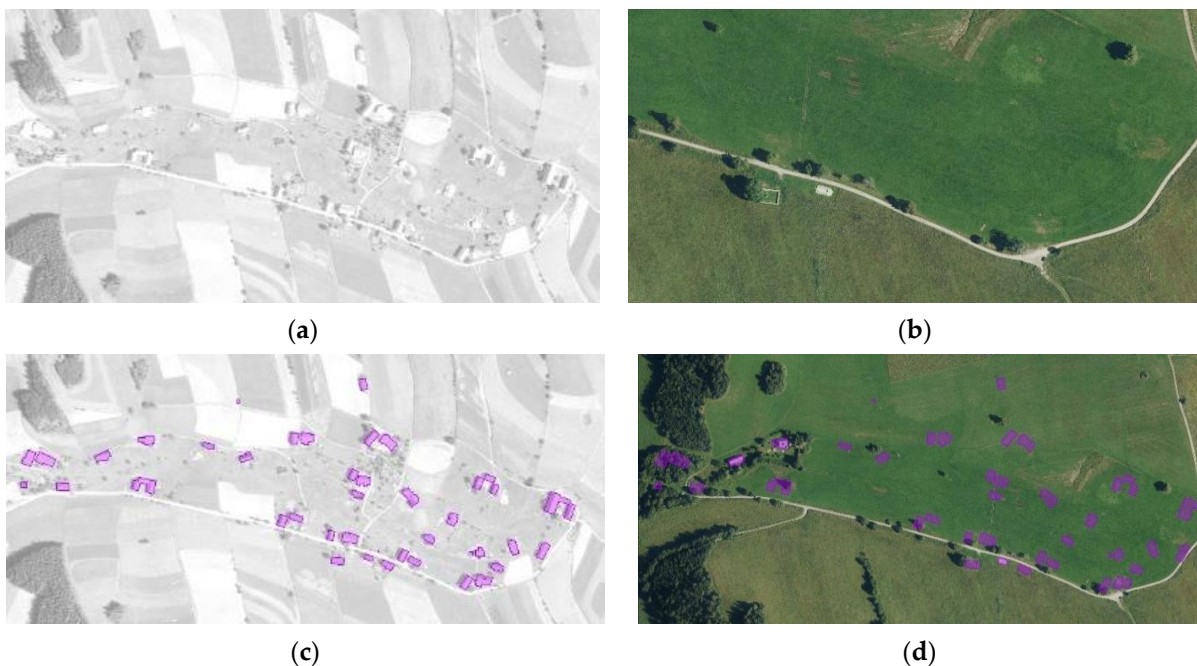

**Figure 3.** Identification of abandoned settlements based on the aerial survey images comparison: (**a**) aerial survey image (1946); (**b**) aerial survey image (2020); (**c**,**d**) marking the demolished buildings' locations. Source: Ortofoto 2020 (© ČÚZK, www.cuzk.cz (accessed on 15 January 2020)), image 1946—VGHMÚř Dobruška, © Ministry of Defense of the CR.

The first step was the analysis of small (dotted), spatial and linear structures on historical aerial surveying images and their comparison with current orthophotomaps, lidar scanning images and drone images. This was followed by a targeted field survey combined with a broader survey of the entire cadastre territory of the abandoned settlement, aimed at determining the more minor features that could not be read from the aforementioned documents. The broader survey took place mainly in the vicinity of the extinct buildings, along the extinct roads and formerly cultivated and (currently) abandoned agricultural land. The visibility of the features identified by the wider field survey on the above-mentioned materials (orthophoto, drone survey, lidar scan) was subsequently checked retrospectively. See Figure 4 below for more details. Individual treasure trove sources of information on extant historic features were evaluated for accessibility, interpretive reliability and added informational value. Based on this evaluation, a recommended procedure for the identification of historic landscape features in areas of radical land use change was compiled.

The specification of the scale category of elements:

- Small structure: several $dm^2$—max. 5 $m^2$;
- Linear structure: the decisive factor is the elongated shape of the element, which is surrounded on both sides by a different environment; min. length of the line is 1 m, but each structure has to be assessed individually;
- Spatial structure: min. 5 $m^2$ (each structure has to be assessed individually).

A detailed diagram of the methodological procedure is presented in Figure 5.

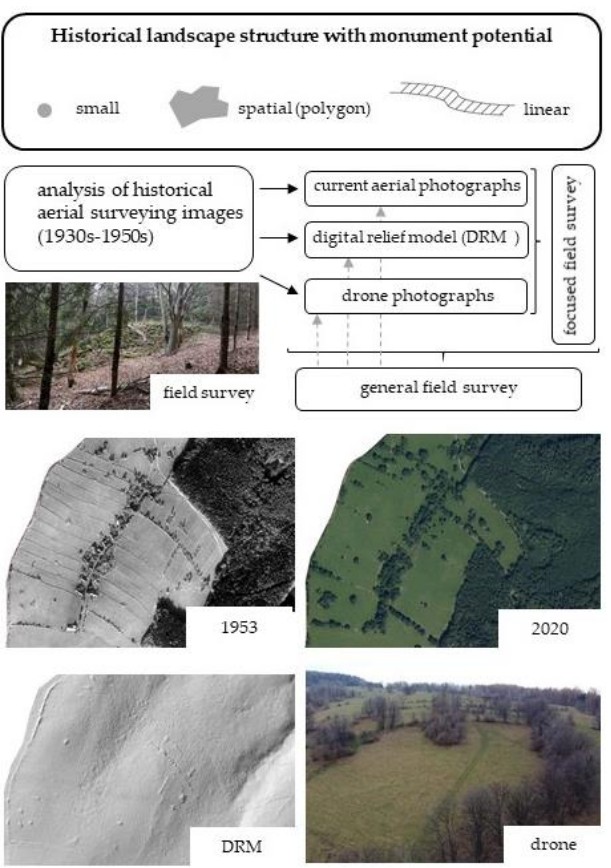

**Figure 4.** Scheme of methodical procedure.

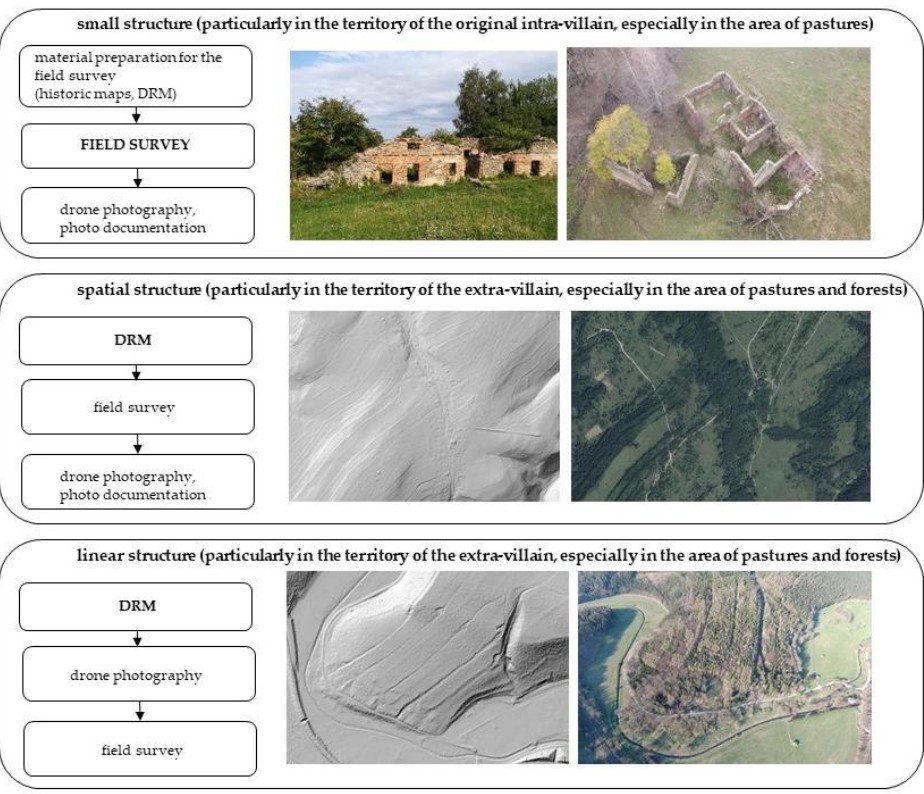

**Figure 5.** Diagram of the methodical framework for the study of the historic landscape structure in localities of abandoned settlements.

## 3. Results

### 3.1. Basic Context—Overview of Identified Historical Landscape Features

A total of 51 sites located in close proximity to the state border was evaluated. The altitude of the sites ranged from lowland areas of 250 m to foothill areas at an altitude of about 700 m above sea level. In terms of typology, the features listed in Table 1 can be identified at the sites in question on the basis of the overall methodology (a combination of all the above steps).

**Table 1.** Typology of features identified on the territory of the extinct settlements.

| Type | Specification | Character |
| --- | --- | --- |
| small (dotted) | surface localisation | agrarian heaps, heaps of building stones, ruins of buildings, bridges, small stone walls; old fruit trees, deciduous solitary trees |
| | subsurface localisation | cellars, wells |
| linear | elements linked to the road network | historical paths (e.g., original stone paving, many bollards), alleys |
| | elements related to management | agrarian bunds, terrace farming; linear greenery (excluding avenues) |
| spatial | interconnected network of elements | preserved landscape texture, preserved structure of the plain (previous farmhand), road network |
| | integral territory | building stone quarries, building plans, cemeteries |

### 3.2. Trends of Landscape Structure and Texture Changes

Changes in the landscape structure and the long-term land use on the sites of the disappeared settlements in Moravia and Silesia after the Second World War correspond to the Czech trend of afforestation at higher elevations and increasing the area of soil blocks. Due to the peripheral location of all the monitored sites, it is possible to observe a long-term stabilised landscape structure, and, at the same time, it is possible to determine these trends of landscape structure changes:

- Afforestation of open visual sites;
- Radical transformation of the structure of agricultural land stock without historic landscape structures;
- Transformation of the structure of agricultural land stock with a significantly preserved historic landscape texture.

Within the field survey, the analysis of drone images and the digital relief model, it was possible to distinguish the intra-villan from the extra-villan of the village on the basis of the partially preserved road network (denser network in the central part of the original village), the newly created clusters of scattered greenery in the places of the original buildings, or, on the contrary, the non-forestation of the original areas of the intra-villan with the simultaneous afforestation/spontaneous expansion of woody vegetation in the surroundings.

Almost all sites were economically exploited. The type of exploitation depended on the altitude in the following gradient, from the lowest to the highest positions: intensive crop production, extensive pastures, and monoculture forestry.

The change in landscape texture is very pronounced in the monitored sites. Almost all sites experienced a partial loss of historical landscape texture (see below for details), but very often, the basic skeleton of the original landscape structures was preserved. The reason for this change is the consolidation of agricultural land and intensive forestry. A

visual comparison of landscape changes in the example of the extinct village of Pelhřimovy is shown in Figure 6.

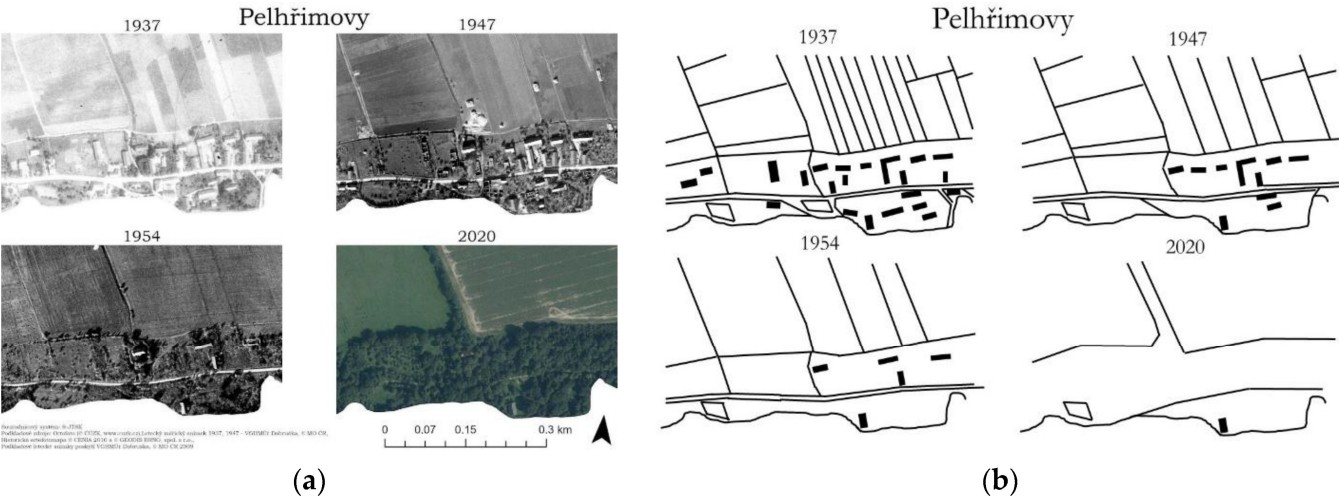

**Figure 6.** Graphic materials for a comparative presentation of the dynamics of (**a**) landscape structure; (**b**) landscape texture. Source: Ortofoto 2020 (© ČÚZK, www.cuzk.cz), image 1946—VGHMÚř Dobruška, © Ministry of Defense of the CR.

*3.3. Spatial Characteristics of the Identified Landscape Features*

It was possible to identify at least one original settlement relict on the territory of each abandoned settlement. The best observable element is the ruins of buildings, which, however, only appear in about a third of the locations. Changes in the relief indicating the original intra-village can be commonly observed in the DRM (up to 90% of the monitored locations). The least widespread element is relicts in the form of underground buildings and historically paved roads. Native trees (fruit trees or other deciduous trees) can be identified quite often.

In the intra-villan of the village, there is only a minimal number of original elements proving the original settlement. As a rule, these are the ruins of buildings or the piles of building stone at the sites of the original building plots. These remnants of building material are typical for almost all extinct settlements. The material shows variability in terms of material used and original purpose. Most commonly found here are phyllite, gneiss and slate due to the regional geology. In most cases—due to the massive removal of original material for use in new buildings in other locations—these are smaller structures in the footprints of former building plots or near roads. In addition to the stones that once formed the outer walls of the original buildings, the sites often contain the remains of roofing materials in the form of slate sheets with typical holes for anchoring them onto the roof structure.

The ruins of the original buildings are of unique value, providing very valuable evidence of the past settlement of these remote localities as well as the materials used, construction techniques and settlement in often very difficult climatic conditions. The sites assessed can be divided into settlements in terms of the preservation of original buildings:

- With few preserved original buildings, usually outside the central part of the original village or settlement;
- With ruins illustrating the specific genius loci of the area (Figure 7);
- Without remains of the previous settlement.

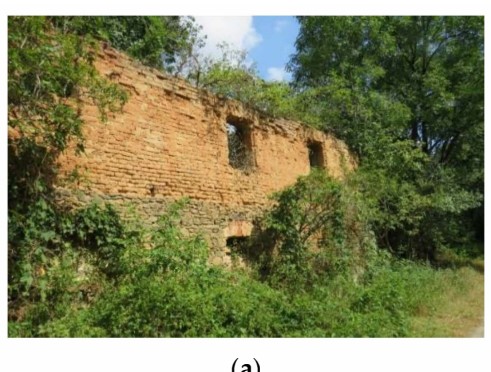 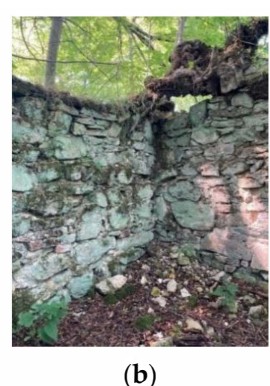 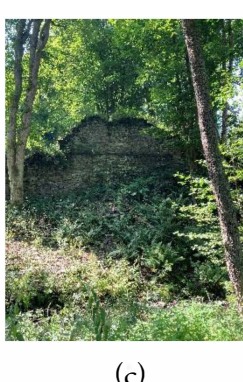

(**a**)  (**b**)  (**c**)

**Figure 7.** Ruins of buildings: (**a**) Pelhřimovy 50.1784936 N, 17.6598617 E; (**b**) Hraničky 50.3106700 N, 16.9746497 E; (**c**) Libavá 49.7012961 N, 17.5890353 E.

In the selected sites, sacral buildings were also preserved (nine sites); in this context, no significant dependence on the current use and altitude could be traced. The spatial differentiation and the precision of the demolition work, which depends on the local administration, is more evident here (also refers to the demolition of ordinary buildings), with the largest number of such buildings in the eastern part of Bohemian Silesia.

Less frequently, non-forest vegetation, especially original solitary deciduous trees (most often lime and ash), is preserved in the original intra-villages. The location of these elements is typically in the immediate vicinity of the disappeared buildings (also confirmed by Majewska [43]). Historic tree plantations and avenues without current direct connection to the road network can be identified in the territory of abandoned settlements. On historical maps, it is possible to trace the roads to which these elements belonged in the past. However, the occurrence of these features is rather rare in the study area.

Old fruit trees are another visible sign of past permanent settlement on the sites of vanished settlements. These elements are evidence of the orchard and fruit-growing tradition in the region and create the potential for their renewal. Fruit trees used to be a common feature of fields, gardens, meadows and pastures in the Czech–Polish border region, but today they sporadically complement the coarse-grained landscape mosaic, mostly with mono-functional use. Solitary old fruit trees can be found both in open landscapes as part of meadows and pastures and as part of today's woodlands. Old fruit trees make an important contribution to the specific historical and landscape footprint of the cultural landscape.

The elements identified in the village intra-villan can best be identified by a combination of detailed field surveys and drone imagery, especially because the exclusive preservation of point microstructures and area and linear elements could not be regenerated due to continuous relatively intensive farming. Surface point features are traceable in historical mapping. These are mainly solitary trees in the vicinity of the original buildings. This step can, therefore, only be considered complementary in order to confirm the location of the tree at the original building and to confirm its historical origin. The other point features that are abundant in the area are more likely to be subsurface structures that can only be identified by field surveys (wells, cellars, but also bridges and other structures). The same applies to small-scale surface structures (bordering on point structures)—these are mainly preserved ground plans of original buildings (clear levelling of the terrain, including any surrounding slope modifications that correspond to the original location of the building). Visible building footprints are preserved exclusively in the current woodland.

The territory of the vast majority of the original village intra-villan remained open in space (it is not forested; only scattered greenery is present). The predominant use of these areas is extensive grazing (intensive agriculture is typical only for the rare lowland areas). In the other areas (5 sites in total), there has been targeted afforestation of the site, including the original intramural area. It can be concluded that the current land use does

not have a significant impact on the preservation of historical landscape elements in the original areas of intra-villans.

The open landscape adjacent to the formerly built-up part of the settlement is characterised by spatial and linear structures, which are preserved to a greater extent in the higher locations. However, point structures can also be found—typically in the form of agrarian mounds (a dome-shaped anthropogenic landform made up of stones loosely stacked on top of each other; these are smaller formations of a non-linear nature associated with agricultural farming—see below). These structures are best identified in the DRM (digital relief model). This is due to their location in the current forest cover (the former use of these areas was demonstrably agricultural—in the form of arable land). A combination with field surveys is ideal to confirm the type and physical form of the structure. In the field, these features often blend in with the surrounding vegetation and residual wood piles (Figure 8). Supplemental drone imagery may be used; it is recommended in areas with no vegetation cover.

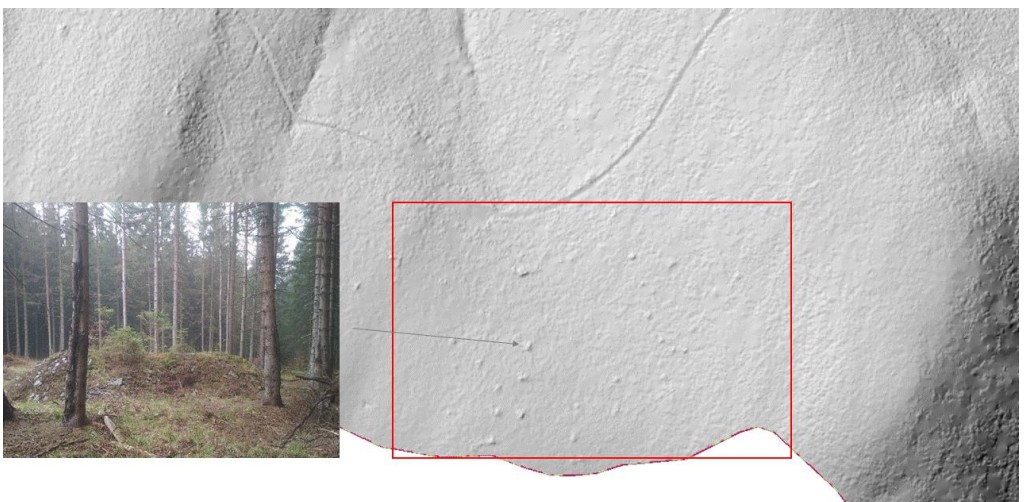

**Figure 8.** Stone pile—demonstration in the field and in the DRM, elements can be observed in the red frame; ZABAGED®Height chart DMR G5 (ags.cuzk.cz); adjusted.

A related typical linear element of the higher locations of the extinct settlements is agrarian mounds (stone walls). These features provide evidence of previous agricultural use of the landscape and are often the only reminder of the former daily presence of people in these remote locations. These structures are typically located outside the original building plots on the edges of previously farmed land (ploughland). These features are typical of sloping land with shallow stony soil. In order to increase fertility and improve soil cultivation, stones were collected and loosely deposited on the edge of the land, where they formed a natural boundary and had an anti-erosion function.

Individual sites are highly variable in terms of the shapes of stone structures. The most frequent are long stone walls (mounds) in open landscapes; exceptionally, they may contain niches (findings of such stone walls on the Polish side are confirmed by Latocha) [44]. Sites with stone walls are currently most often used as pastures. However, stone walls (Figure 9) can also be found within forest stands. The woodland is generally typical of the highest elevations and very steep slopes, where agricultural management would be unthinkable today.

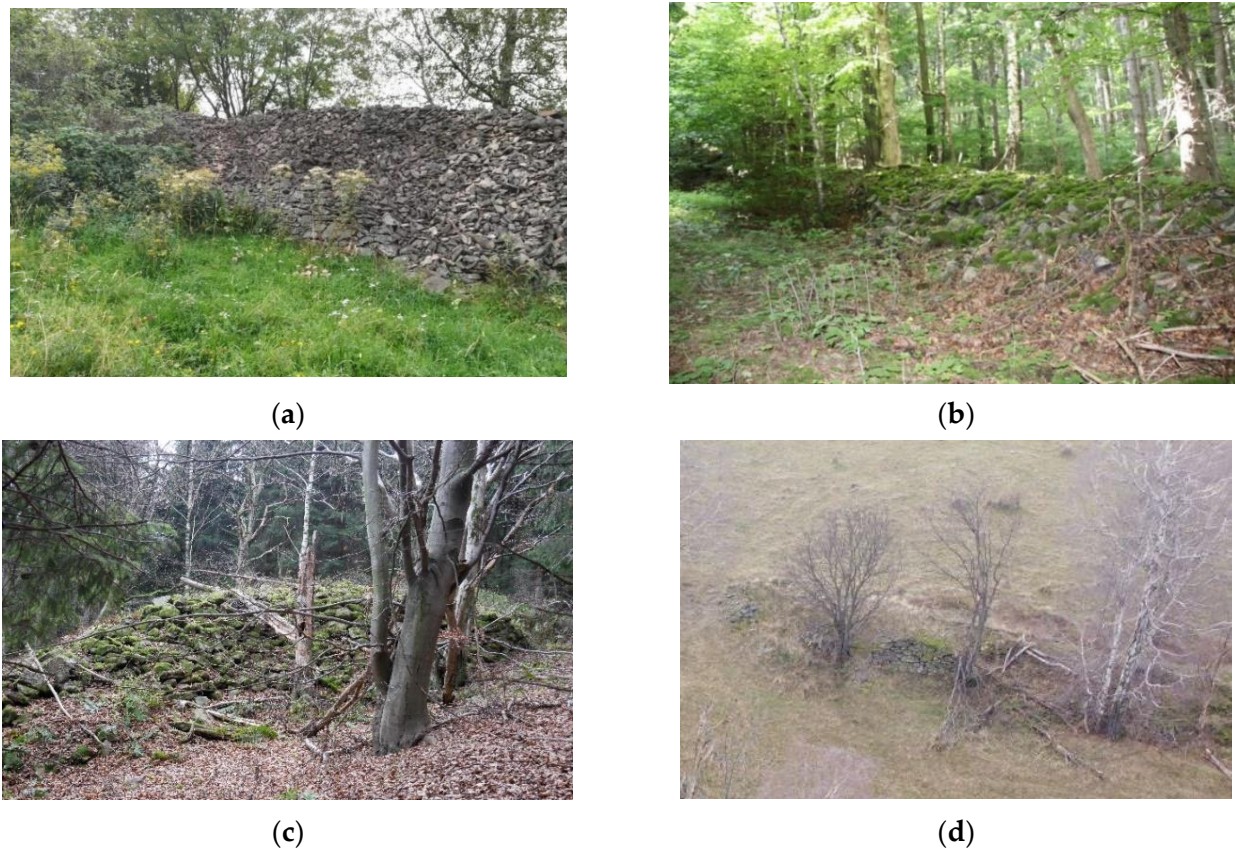

**Figure 9.** Stone walls: (**a**) Hřibová 50.3468017 N, 17.0068733 E; (**b**) Kamenné 50.2874683 N, 17.0471761 E; (**c**) Hřibová 50.3465264 N, 17.0137519 E; (**d**) Hřibová 50.3478356 N, 17.0129886 E.

Stone structures add to the character of the contemporary landscape and support the specific genius loci of abandoned sites. Apart from their aesthetic dimension, they contribute significantly to the biodiversity and stability of the current landscape system (e.g., [45–47]). The question of formalising their protection in Czech law is very topical.

The location of these features is possible on the basis of current orthophotos, provided that the feature is located in view-open locations and without vegetation line cover. This combination is not very common. However, the linear foliage that often accompanies stone bunds in the form of network structures can be considered an indicator of the occurrence of these anthropogenic landforms. Once identified, a combination of field surveys and drone imagery is necessary. The DRM is the most suitable basis, as it captures structures regardless of the current land use, and it is possible to identify even small linear microstructures in the terrain hidden by dense forest cover. However, a field survey is also required here.

Another relatively common (though partially) preserved element is road networks. With the disappearance of man from the landscape, many of the roads have disappeared, but they are still visible in the present relief—whether they are smaller footpaths or massive bridleways cut into the terrain. In this context, it is necessary to distinguish the actively used road network today, which is preserved only in axial roads. Residual, currently unused historic roads are often not visible in the terrain. However, they can be very well observed in the DRM, where lines corresponding to the historical state of the road network can be identified even in the field. These structures are most often located in forest stands. In Figure 10, we can see a set of historic droveways that were used to manage the fallow (strip) land belonging to individual farmsteads located in the lower parts of the property. Today, this area is covered by woodland.

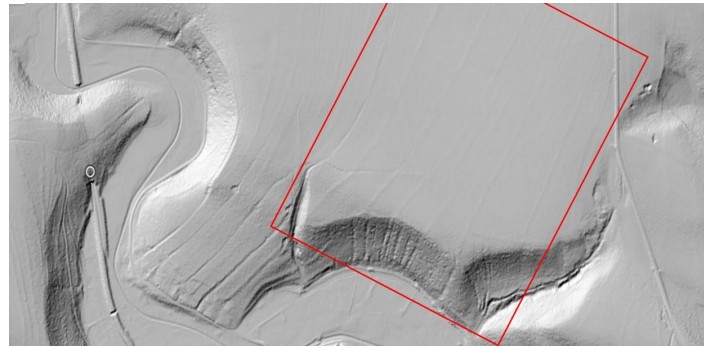

**Figure 10.** System of historic tracks in the DRM, ZABAGED®Height chart DMR G5 (ags.cuzk.cz, (accessed on 8 May 2012)., accessed on 20 July 2022); adjusted.

A related surviving historical landscape element of a flat character is the preserved ploughland (a set of former fields with a different current use, roads, agrarian mounds or terraces); see Figure 11. This flat structure is preserved in most of the higher locations, regardless of the type of farming (grazing, forestry). This structure is most visible in the DRM. A combination with field surveys is possible, which has a verification but not identification character (in the field, these structures are often not visible at all in a contextual view but rather as separate microstructures). The other sources of information are only applicable in the case of plastic structures (stone bunds and terrace farming); however, the image of the original pluvium is only partially visible compared to the images in the DRM. This output provides a very valuable record of the structure of agricultural land in the past.

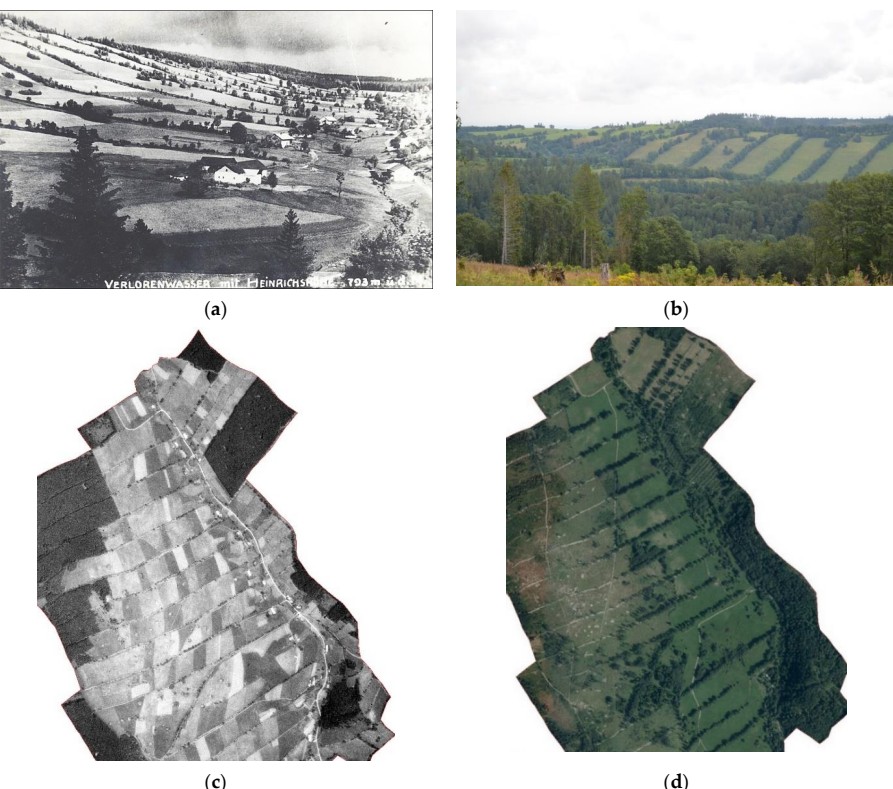

**Figure 11.** Preserved landscape texture (Ztracená Voda, 50.2030528 N, 17.4801681 E): (**a**) photo from the beginning of the 20th century; (**b**) actual photo (2020); (**c**) aerial survey image (1937); (**d**) aerial survey image (2020). Source: Ortofoto 2020 (© ČÚZK, www.cuzk.cz, accessed on 20 March 2020), image 1946—VGHMÚř Dobruška, © Ministry of Defense of the CR.

Local sources of building material in the form of small-scale quarries can also be found in the vicinity of the extinct settlements as part of the wider hinterland of individual estates. According to witnesses, these quarries served as a repository of material for repairing roads, houses, walls, etc. However, these structures are poorly observable in all types of evidence, including field surveys, and have mostly been incorporated as part of demolitions and controlled landscaping or have blended into the surrounding area through natural succession.

Sporadic surviving features are also represented in the sites by the torsos of cemeteries, cobbled paths and stone bollards along defunct roads. In the Czech–Polish border area, these elements are very rare. The first reason is the more precise demolitions in localities near the state border and the long-term isolation of this area; the second reason is the total change of management in the landscape (mono-functional agriculture or forestry combined with land consolidation)

### 3.4. Methods of Identification

Spatial and linear historic structures are better preserved (more frequently and to a greater extent) in areas that have been reforested compared to areas that are used for extensive grazing and particularly for crop production.

Most point line elements are located in the places of the original intra-village, whose territory is currently used as pasture. A field survey is necessary for identification; other documents can be used to prepare a targeted survey and local drone photography.

The other structures—linear and space (spatial)—are best observed in the extra-village, with simultaneous use for forestry and grazing. The primary basis for the detention of these structures is the DRM.

For a general overview, see Table 2.

**Table 2.** Overview of the occurrence of landscape structure types according to current land use.

| Localisation/Type | Intra-Villan Arable Land | Extra-Villan Arable Land | Intra-Villan Pasture | Extra-Villan Pasture | Intra-Villan Forest | Extra-Villan Forest | Source |
|---|---|---|---|---|---|---|---|
| small | | | | | | | survey |
| spatial | | | | | | | DRM |
| linear | | | | | | | DRM |
| preservation of landscape structures | | | | | rare | partly | significant |

All spatial and linear features identified in the DRM were verified, but a combination with a targeted field survey was required. However, this survey could not reveal the overall composition (e.g., network, connected pelements) but only separate sub-parts. On the other hand, no false negative results were detected—all plastic linear and spatial elements could be observed to the same extent in the DRM as well as within the field survey.

## 4. Discussion and Conclusions

The tendency to abandon landscapes and settlements is nothing new. What is new is the optics of looking at this phenomenon. Previously, the abandonment of landscapes was viewed negatively or as a 'banalization of the landscape' [48]. Today, the optics are directed towards nature conservation combined with extensive small-scale agriculture. This shift is also confirmed by D'Angelo [49]: the current trend can be described as a return to the appreciation of traditional agricultural landscapes—mostly for biodiversity conservation reasons, but also for cultural and historical motives. Human activity is not in conflict with biodiversity. Strengthening the links between biodiversity conservation and grassland maintenance/restoration (including rural built heritage) is not only an opportunity but probably the only way to preserve these unique places in the long term [50]. Failure to respect native landscape structures, including the driving forces that enable their creation or protection, hinders the enhancement of landscape and biocultural diversity and the positive integration of socio-cultural and environmental diversity in general [51].

From the point of view of historical landscape structures, the linear and surface structures that are located outside the original intra-villan of the municipality are significant for the border localities of extinct settlements. The preservation of these structures can be expected to a greater extent in areas with formal landscape protection, a trend confirmed by Sklenička [52]. However, the assessed areas are usually located almost entirely outside the territorial nature protection guaranteed by law. Thus, the preservation of structures can be attributed mainly to higher elevations and extensive agriculture or to the preservation of these structures in forest cover (this is especially the case for historic ploughland). Subsidies also play a role, particularly in relation to grazing at higher altitudes. Research by Aimar [53] confirms the need for the active conservation and management of historic landscape structures as indicators of landscape integrity and quality, a prerequisite for their use in place-based landscape management. This is related to the change in the current paradigm of the issue addressed, which is shifting from the simple conservation of values to their creative use in local development [54]. The need to introduce new practices based on the compatibility between conceptual human action and biodiversity enhancement (with emphasis on peripheral areas of mountain and foothill landscapes), which will result in a stabilised landscape with socio-cultural potential for local populations, is also highlighted by García-Ruiz et al. [55].

All identified elements contribute significantly to the diversity of the cultural landscape, especially in terms of recording the historical land use of the territory and the long-term time required for their creation. Moreover, these elements were preserved without special protection. This testifies to the permanence of the human footprint on landscapes. This applies in particular to relief traces—e.g., visible floor plans of buildings, ploughs, driveways, etc. These elements also usually create habitats for other species of plants and animals. Hence, we can talk about strengthening biodiversity on several levels with a mutual effect (e.g., [56,57]). Scherreiks et al. [58] came to the conclusion that species richness cannot be unequivocally explained only by the current conditions of the landscape and that the historical structure of the landscape is relevant for the high species richness observed today. This thesis is also the reason for the registration and protection of historical landscape elements [59].

The typology of historical landscapes for Czechia was prepared by Erlich et al. [60]. They divided historical landscapes into composed, organic and associative landscapes. In this systematization, the landscape of extinct settlements would mostly fall under organically developed relict landscapes, where evolution has already ended, but significant characteristics persist. The research importance of these historical landscapes (composed not conceptually but on the basis of joint use and cultivation of land) is confirmed by Kučera et al. [61].

The paper proposes a methodology for the analysis of landscape elements in the area of extinct settlements and their possible typology (and their specific forms) on the example of abandoned settlements in the eastern part of the Czech–Polish border area. This methodology is also applicable to other Central European territories after possible modification according to local conditions. Its purpose is its potential use for landscape planning and other decision-making processes affecting the landscape. The methodology uses a combination of old maps and historical information with modern remote sensing methods, including the deployment of drones. It can be assumed that the possibilities of these methods will be further developed. However, it should be stressed that despite the expected advances in modern methods, field surveys and work with historical sources remain an integral part of the methodology.

Climate change poses a certain challenge to the landscape of abandoned settlements. Given the location of these settlements in mountain and foothill areas, the threat of drought is not as urgent here, which could lead to some revitalisation of agricultural production. In this context, the low risk of drought can be understood as an advantage compared to lower locations. Current developments also show the limits of globalisation. The idea of unlimited travel and global cooperation is taking hold. This could lead to a new perspective on the use of domestic land.

Significantly, there is virtually no fallow land in the study sites [62]. The disappearance of settlements did not mean the abandonment of the landscape, which continues to be exploited. Traditional agricultural and forestry uses have been joined in the post-reproductive era by tourism uses.

In this way, the landscapes of vanished settlements in the post-productive period acquire another function—that of tourism. It is possible to use the qualities of the formerly urbanised landscape, returned to a greater or lesser extent to the open landscape, which in mountain and foothill positions acquires the aesthetically positive qualities of a mosaic of forests, meadows, fields, water areas and streams, scattered greenery and remnants of settlements. Educational trails are being built to remind people of the development of the landscape and its causes. Part of this may be nostalgic tourism [63], where former residents or their descendants return to places linked to the history of their lives; this is expressed through emotions referred to as heimweh [64]. If the disappearance of a settlement is linked to violent events or disasters, it could also be dark tourism [65]. Latocha [66] suggested a possible return to agricultural use, the renovation of old houses, the partial restoration of the sacral landscape, and tourist infrastructure and educational initiatives (educational trails, eco-museums, information boards) for the Polish Kłodzko afforestation.

Although the landscapes of the extinct settlements are in relatively good condition, greater tourism development has been hampered by inadequate infrastructure [67]. The area is also specific in that the original settlers were displaced to Germany and very limited access was only allowed after the political liberation. Rather, in the post-war period, the state sought to sever ties with the indigenous population and build new relationships. However, this was clearly not successful in the case of the disappeared settlements. There are hardly any original survivors left. Therefore, the development of the landscape and its use for tourism is actually in its infancy. The dissemination of knowledge and information is a crucial issue in this respect.

The question of cross-border cooperation remains. Given that the Polish side has also undergone processes of post-war population exchange based on ethnicity with similar consequences, such cooperation would be directly offered. On the other hand, there are also differences in relation to the landscape, as there has not been such consistent collectivisation in Poland.

Further physical disappearances of rural settlements in large numbers are not expected. However, the abandonment or disappearance of industrial or infrastructural sites and buildings may have similar consequences for the landscape. Therefore, further monitoring, analysis and assessment of the landscape of abandoned settlements is expected.

**Author Contributions:** Conceptualisation: H.V.; methodology: H.V.; data collection: H.V. and V.P.; validation: A.V.; formal analysis: H.V.; resources: A.V. and H.V.; data curation: H.V. and V.P.; writing—original draft preparation: H.V.; writing—review and editing: H.V.; visualisation: V.P.; supervision: H.V.; project administration: H.V.; funding acquisition: H.V. All authors have read and agreed to the published version of the manuscript.

**Funding:** This research was funded by the Ministry of Culture of the Czech Republic, grant number DG18P02OVV070.

**Institutional Review Board Statement:** Not applicable.

**Informed Consent Statement:** Not applicable.

**Data Availability Statement:** Not applicable.

**Acknowledgments:** This research was realized within the terms of the project titled "Identification and permanent documentation of the cultural, landscape and settlement memory of the municipality—on the example of extinct settlements of Moravia and Silesia" financed by the Ministry of Culture of the Czech Republic under the Program to support applied research and experimental development of national and cultural identity for the 2018–2022 period NAKI (National and Cultural Identity) II, No. DG18P020VV070.

**Conflicts of Interest:** The authors declare no conflict of interest.

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
