# Peer review of "Historical Landscape Elements of Abandoned Foothill Villages—A Case Study of the Historical Territory of Moravia and Silesia"

_land, doi:10.3390/land11101809_

Round 1

Reviewer 1 Report

The paper deals about interesting topic but it must be seriously improved in its presentation.The overall presentation of the topic is low and unclear.

Introduction:
The major part of introduction (L26-71) sounds more like model area description. This part is too long and descriptive, nevertheless, references to the historians books and papers are missing.

On the other hand, L84-155 sound like Introduction, so move this part to Introduction.

The part about settlements abandonment (L84-130) is too long and descriptive, it should be shortened.

On the other hand, in the Introduction should be discussed papers about historical landscape structures, identification of these structures, historical landscape typology, methods of historical cultural landscape identification, concept of landscape heritage. All these topics are highly relevant to the presented work.

Add the chapter model area where the model region will be presented, e.g. natural conditions such as altitude, climate, and administrative units.

Add the chapter Data and present the used data clearly. When the old aerial photographs were taken?

Present methods more deeply. How did you do the first identification of the abandoned villages? Were all identified villages incorporated to the research? Add the table with list of studied old landscape features and their short characterisics. What was the smallest lenght of linear features and area of spatial features? To what extent do you study the surrounding of the abandoned village (how big was the buffer)?

L143: The reference ŠÅ¥astná et al. does not correspond with the number 36.

Fig. 1b: Captions are too small and unreadable.

Results:

The chapter should be divided to sub-chapters (e.g. types of the historical landscape features, methods of identification) and presented in more orginized way.

It will be interesting present in table how many historical landscape features of each type were identified.

Tab. 1: Words "spatial" and "linear" in the left column are switched.

L217-220: Add the reference.

L337-338: Add the reference.

Tab. 2: If I understand good, the cathegories intra-villain and extra-villain on one side and arable land, pasture, forest on the other side could be complementary (e.g. actual arable land in the extra-villain)? Present it more clearly in the table.

Discussion:

Add the discussion about ways and methods of identification of the historical landscape structures, types of historical landscape structures and their good/bad state, present land coved on the historical landscape structures.

On the other hand, don´t discusse problems that are not reflected in the results such as nature conservation (L419-432) and tourism (433-454).

What do you mean under the term "conventional agricultural landscapes"?

Conclusion:

The possible classification (L463) should be presented in the results first.

Move the Fig. 10 to the Methods.

Improve the English language. There are repetitive parts like "sources of mineral resources" (L37).

Author Response

Thank you very much for your useful and helpful comments.

Reviewer 2 Report

The authors present an interesting study of abandoned settlements in the Czech Republic, combining several different types of datasets, including historical imagery, field survey, drone survey, and lidar. I enjoyed the paper and the creative use of data. I have a few comments that I think will improve the paper.

Overall, the paper could be slightly better structured and organized. For example, the Materials and Methods section begins with a more theoretical or literature review section that is very interesting but seems rushed. The authors could elaborate more and move the first half of this section to an earlier introductory section. The authors should also expand on the second half of the section, which actually does belong in the Materials and Methods section. I would like to see more discussion of the source of the lidar data, type of drone used, etc. What was the return density of the lidar, what was the resolution of the DRM -- these metrics can have profound effects on the identification of linear features and standing architecture. Did the authors consider using vegetation returns from the lidar, for example a canopy height model to assess vegetation like old fruit trees and deciduous solitary trees? What year was the lidar data collected in relation to the drone imagery?

The terms landscape structure and landscape texture are used repeatedly, but I would like to see the authors better define this terminology at the outset.

The authors mention several times (line 336 for example) that these cultural resources continue to affect biodiversity, but this point should be elaborated more -- in what ways do these features influence modern biodiversity?

Regarding the field survey -- was this done after analyzing the lidar -- were all features identified in the lidar ground verified? Were there any features identified on the ground that did not appear in the lidar (false negatives)?

The section beginning with line 386 raises many interesting ideas, but these thoughts are expressed essentially as bullet points -- these areas should be expanded and elaborated.

Line 473 raises climate change but immediately rejects the influence of drought; surely there must be other threats posed by climate change on this region?

Finally, as one of the goals of the paper is to present a typology of features, I think this area can be improved. Table 1 is a great start but a bit confusing. I did not understand what is meant by small (dotted) features, and why are historical paths classified as spatial while building plans are linear? That seems counterintuitive to me. Please elaborate more on this typology and define terms like small, spatial, and linear, and why certain features were assigned to these categories.

Author Response

Thank you very much for your useful and helpful comments. We are sending the reactions in enclosure.
